# Bit-by-Bit: Investigating the Vulnerabilities of Binary Neural Networks to Adversarial Bit Flipping

**Shamik Kundu**[*1]**, Sanjay Das**[*1]**, Sayar Karmakar**[2]**, Arnab Raha**[3]**, Souvik Kundu**[3]**, Yiorgos Makris**[1]**, Kanad Basu**[1]

[1]*University of Texas at Dallas,* [2]*University of Florida,* [3]*Intel Corporation*

## Abstract

Binary Neural Networks (BNNs), operating with ultra-low precision weights, incur a significant reduction in storage and compute cost compared to the traditional Deep Neural Networks (DNNs). However, vulnerability of such models against various hardware attacks are yet to be fully unveiled. Towards understanding the potential threat imposed on such highly efficient models, in this paper, we explore a novel adversarial attack paradigm pertaining to BNNs. In specific, we assume the attack to be executed during deployment phase, prior to inference, to achieve malicious intentions, via manipulation of accessible network parameters. We aim to accomplish a graceless degradation in BNN accuracy to a point, where the fully functional network can behave as a random output generator at best, thus subverting the confidence in the system. To this end, we propose an Outlier Gradient-based Evolutionary (OGE) attack, that learns injection of minimal amount of critical bit flips in the pre-trained binary network weights, to introduce classification errors in the inference execution. To the best of our knowledge, this is the first work that leverages the outlier gradient weights to orchestrate a hardware-based bit-flip attack, that is highly effective against the typically resilient low-quantization BNNs. Exhaustive evaluations on popular image recognition datasets including Fashion-MNIST, CIFAR10, GTSRB, and ImageNet demonstrate that, OGE can drop up to 68.1% of the test images mis-classification, by flipping as little as 150 binary weights, out of 10.3 millions in a BNN architecture. Code is open sourced at: `https://github.com/isnadnr/OGE`.

## 1 Introduction

A commitment to reducing size and compute demands has led to ultra-low-precision BNNs, featuring one-bit weights and activations (-1 or +1). Introduced by Courbariaux et al. (2016), BNNs drastically improve power efficiency and inference latency by minimizing memory access and using fast bit-wise operations instead of complex matrix multiplications. These improvements maintain classification accuracy compared to high-precision Deep Neural Networks (DNNs) (Qin et al., 2020; Yuan & Agaian, 2021). BNNs are favored for Machine Learning as a Service (MLaaS) across hardware platforms (Sanyal et al., 2018). Recent advancements in Deep learning have integrated low-precision DNNs into critical domains like facial recognition (Dong et al., 2019) and autonomous driving (Eykholt et al., 2018).

However, this widespread adoption of DNNs has ushered in a concerning surge in adversarial attempts, exploiting network vulnerabilities through backdoor and inference attacks (Saha et al., 2020; Xie et al., 2019; Goodfellow et al., 2014; Moosavi-Dezfooli et al., 2017). In contrast to these known attack vectors, there exists a relatively uncharted territory: an innovative attack paradigm centered on the manipulation of pre-trained model weights (Breier et al., 2018). Executing such an attack hinges on a rather menacing threat model, assuming that the adversary possesses unrestricted access to a device's memory, enabling direct parameter alterations within a deployed model to serve adversarial objectives. Given that the deployed DNN is stored

---

[*]Authors have equal contributions.

This work is supported by the Semiconductor Research Corporation (SRC GRC Task: 3243.001).

in a binarized format in the memory, attackers can tamper with model parameters employing techniques like the Row-Hammer Attack (Agoyan et al., 2010; Kim et al., 2014b) and the Laser Beam Attack (Selmke et al., 2015), as illustrated in Figure 1. While bit-flip attacks have demonstrated their potential to wreak havoc at the system level, those targeting the control path are comparatively easier to address, as they are integral to the overall system's integrity. Conversely, datapath attacks, which surreptitiously manipulate accuracy, operate in stealth mode, posing a significant challenge in detection. An instance of misclassification during inference may be erroneously dismissed as an intrinsic network characteristic, when it is, in fact, a consequence of a stealthy bit-flip attack.

Existing research have demonstrated that it is feasible to change the model weights via bit flipping to accomplish malicious intentions (Rakin et al., 2019; 2020; 2022; Bai et al., 2021). However, these techniques are either based on heuristic strategies (Rakin et al., 2019), or focused on identifying critical weights in high precision DNNs (Bai et al., 2021). Since BNNs are emerging as promising candidates to be deployed in security-critical high assurance environments, exploring the vulnerability of such low precision networks is imperative to estimate its ad-

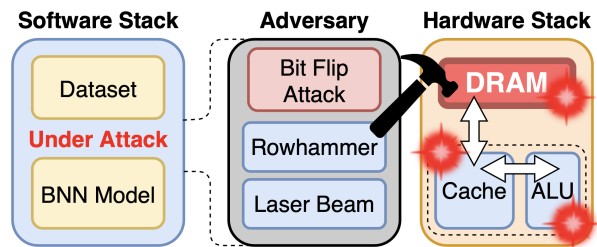

Figure 1: Threat model demonstrating Bit Flip attack.

versarial impacts. These BNNs are generally apprehended to be resilient against bit flip attacks owing to their limited parameter space (Rakin et al., 2021). This resiliency may be attributed to the fact that the magnitude of error by flipping a bit in a BNN is minimal. Since the massive degradation is inherently circumvented in case of BNNs, existing attack strategies that exploit the large bit-width of the weights to orchestrate the attack are not effective in case of these ultra low-precision binary weights in a BNN. Such inherent tolerance can only be disrupted with a significant number of bit flips (usually, over $39\times$, compared to a DNN, as we have demonstrated in this paper) (Rakin et al., 2019; 2021). However, manipulating a plethora of bits disrupts the stealthiness of the attack. As a result, attacking such extremely low precision networks becomes particularly challenging.[1]

**Our Contributions:** In this paper, we challenge the conventional wisdom of treating BNNs to be inherently robust against malicious bit-flips. we devise a technique to determine a diminutive set of most vulnerable binary weights in the BNN, flipping which furnishes a significant reduction in accuracy to a point where the network is deemed as a random output generator. In this direction, we propose a novel Outlier Gradient-based Evolutionary (OGE) attack framework, that systematically identifies the most vulnerable set of binary weights to be attacked in a pre-trained BNN. To this end, we reformulate the task of obtaining the most vulnerable layers in the BNN by ranking the network loss for top-$k$ gradient bit flips in each layer. Subsequently, the outlier gradients from the vulnerable layers are isolated to be provided as input to an evolutionary algorithm. This accomplishes the search space optimization using an entropy-based objective loss function in iterations. Optimization of this loss function caters to the drop in inference accuracy of the BNN. In our proposed OGE framework, the subset selection of outlier gradient weights aids in circumventing the challenge posed while attacking a BNN, by enabling the evolutionary algorithm to obtain the critical weights from this vulnerable search space. To the best of our knowledge, this is the first work that systematically explores the vulnerability in a BNN via adversarial bit flipping, instead of a heuristic strategy. This paper specifically makes the following key contributions:

- We, for the first time, have discovered the significance of outlier gradient weights in a binary context and linked their influence to substantial reductions in classification accuracy within a Binary Neural Network (BNN) due to bit flips.

- Based on the understanding from the outlier gradients, we propose an Outlier Gradient-based Evolutionary (OGE) attack, a novel bit-flip attack framework, hat utilizes an evolutionary search approach,

---

[1]As the weights in a BNN are binarized to be represented using a single bit ($+1$ or $-1$), flipping a bit corresponds to a weight flip in the network. Hence, we use both these terms interchangeably in the remaining paper.

coupled with an isolation forest-based outlier detection framework for furnishing the most critical set of binary weights in a BNN.

- We perform extensive experiments on complex image recognition datasets including ImageNet, and evaluate the efficiency of our OGE attack on both binary convolution neural networks and binarized vision transformers. Our results demonstrate that up to 68.1% of the test images could be misclassified to an incorrect class, by flipping as little as 150 binary weights, out of 10.3 millions in the BNN architecture, which is almost 84% less than the state-of-the-art bit flip attack.

## 2 Background & Related Works

### 2.1 Binary Neural Networks (BNNs)

BNNs are gaining prominence as highly promising candidates for deployment in resource-constrained edge devices, primarily due to their exceptional efficiency in terms of storage and computation. Unlike full-precision networks that demand a 32-bit floating-point representation, BNNs operate with one-bit weights and neuron activations, taking the form of -1 or +1. To shed light on the inner workings of this innovative approach, consider the binarization function employed in BinaryNet, a state-of-the-art binary convolutional neural network as introduced by Courbariaux et al. (2016). This function meticulously considers the sign bit of the real-valued variable, and its representation can be articulated as follows:

$$x^b = sign(x) = \begin{cases} +1 & \text{if } x \geq 0 \\ -1 & otherwise \end{cases}$$

where $x^b$ is the binarized variable (weight or activation) and $x$ the real-valued variable. This function, which binarizes network parameters into -1 and +1, paves the way for a significant departure from traditional compute-intensive multiplication operations. Instead, it replaces these operations with a simple one-bit XNOR operation, followed by a simple bit-counting operation for accumulation. Consequently, the Multiply-Accumulate (MAC) operation, which is highly compute intensive in the realm of neural networks, is executed with remarkable efficiency in Binary Neural Networks (BNNs). This efficiency stands in stark contrast to the computationally demanding floating-point operations prevalent in full-precision Deep Neural Networks (DNNs). In addition to these computational advantages, BNN also delivers substantial energy savings and remarkable speedups, with reported improvements of up to $7\times$ (Courbariaux et al., 2016). Remarkably, these performance gains do not come at the cost of classification accuracy, as demonstrated by various studies (Yuan & Agaian, 2021). These combined attributes position BNN as an optimal solution for edge-based inference tasks, where the trifecta of efficiency, speed, and accuracy is of paramount importance. Furthermore, in addition to the computational and efficiency advantages, the inherent reduced precision in BNNs contributes to enhancing the network's resilience. To illustrate, a weight of $+1$ can eventuate to $-1$, or vice-versa, thereby furnishing a maximum error magnitude of 2. On the other hand, in case of a high precision DNN, *e.g.*, an 8-bit quantized network, a bit flip in the most significant bit (sign bit) can manipulate a specific weight value of $+123$ to $-5$, thus engendering a massive error magnitude. This resilience extends over and above that of traditional high-precision DNNs, especially when faced with attacks aimed at injecting errors by tampering with the network parameters (as we demonstrate later in this paper).

### 2.2 Bit-flip Attacks

Memory bit-flip attacks, often executed through techniques like the Row-Hammer Attack, have garnered recognition as an established threat model. This threat has been extensively investigated and documented in existing literature, as exemplified by Kim et al. (2014a); Razavi et al. (2016). Given that the pre-trained network parameters of a Deep Neural Network (DNN) reside in memory before deployment at the edge, the occurrence of such bit-flip attacks raises legitimate concerns and presents substantial challenges concerning the security and reliability of DNNs. This threat becomes even more pressing in security-critical edge applications, where implementing sophisticated defense mechanisms becomes a formidable task due to resource limitations.

Within the existing body of literature, researchers have explored various methods for injecting faults in neural networks, targeting elements such as biases, weights, and activation functions, with the ultimate aim of inducing both targeted and untargeted misclassifications during the inference process. It's worth noting that these prior techniques were primarily designed to attack full-precision Deep Neural Network (DNN) models. At the edge, however, many DNN implementations operate in quantized precision, as emphasized by Wu et al. (2016), rendering them inherently robust to variations in network parameters. In response to this challenge, efficient algorithms have been developed and put into practice to identify the most vulnerable bits within a quantized DNN, as demonstrated in Rakin et al. (2019; 2022); Bai et al. (2021). The Bit Flip Attack (BFA) utilizes a heuristic progressive search strategy, wherein network layers are iteratively examined to pinpoint the layer that contributes the most to network loss for a certain number of bit flips during each iteration, as elucidated in Rakin et al. (2019). Subsequently, the selected bits are flipped within the most influential layer, and the model is updated, thereby undermining network accuracy. Targeted-BFA (TBFA) represents an extension of the BFA technique, aiming to manipulate data samples with the intent of targeting specific classes, employing the same progressive search approach, as documented in Rakin et al. (2022); Bai et al. (2021).

It's important to emphasize that the bit flip attacks discussed earlier primarily employ heuristic strategies and are tailored for high-precision Deep Neural Networks (DNNs). These attacks indeed rely on exploiting computations that involve network parameters with high-precision values to achieve their intended goals, such as manipulating the network's behavior and causing misclassifications. However, it's worth noting that these attacks are not effective when directed at low-precision binary network weights, which operate using only one-bit values, as opposed to the multi-bit values present in high-precision DNNs. The unique characteristics of BNNs, including their limited precision and binary parameters, make them less susceptible to the types of attacks that target high-precision DNNs. While few recent research has focused on mitigating adversarial noise at the input of BNNs by leveraging characteristic hardware attributes, the impact of bit flip-based attacks has largely been ignored (Bhattacharjee & Panda, 2020; Kim et al., 2022).

## 3 Proposed OGE Attack Framework

In this section, we begin by defining the threat model, which outlines the standard assumptions regarding the capabilities of the adversary. We then propose a novel bit flip attack framework, pertaining to BNNs, that furnish the most critical binary weights in the pre-trained network, which when flipped, cause significant degradation in classification accuracy.

### 3.1 Threat Model

Within the realm of Machine Learning as a Service (MLaaS), individuals upload their DNN models to a platform that may carry inherent security risks due to the desire for increased computational resources. In this setting, DNN inferences and other applications, which may be under the control of potential attackers, often share server hardware components, such as the last-level cache and main memory. While attackers lack explicit permission to read or write data within the user's memory, the existing tools empower them to exploit side channels, thereby gaining access to sensitive information. This discussion categorizes threats into three levels based on the extent of knowledge that attackers can extract from the victim.

#### 3.1.1 Full Knowledge Attack

A full-knowledge attack represents the most critical scenario. In this scenario, attackers can harness powerful tools, such as Cache Side-Channel Attacks targeting the Last-Level Cache (LLC), as detailed in Yarom & Falkner (2014); Kayaalp et al. (2016), to retrieve memory data stored in the cache. Although the throughput of such a channel is limited (less than 1 MB/s as noted in Hong et al. (2018)), it poses a significant risk to the model, especially if sensitive data within the memory is exposed. In this context, we assume that the model's weights may become vulnerable to the attackers. Moreover, attackers are well-informed about the proposed defense methods, like randomized rotation, and have the capability to enhance their attack strategies if any leakage persists. Given the impracticality of a full-knowledge attack due to the limited throughput of this channel, it remains a severe but technically challenging threat.

### 3.1.2 White Box Attack

The attackers lack the ability to directly access data within the memory, but they can make educated estimations regarding the locations of the most susceptible bits, allowing them to potentially execute rowhammer attacks. There are two primary reasons for this: 1) The majority of commercial models and datasets are open source, and model users often download pre-trained models, customizing them through transfer learning (Marcelino, 2018). These customized models inherit vulnerabilities from the open-source models, effectively providing attackers with a source of information. 2) Attackers can employ hardware tools to monitor memory access patterns, as documented in Hu et al. (2020); Yan et al. (2020), enabling them to accurately deduce the model's architecture. Armed with this knowledge, attackers can locally train their own models, which would also share the same vulnerabilities as the user's model. This white-box attack approach is inherently perilous and practical, and it serves as the baseline attack model for the rest of this paper.

### 3.1.3 Black Box Attack

When attackers are unable to acquire any knowledge about the model, an alternative approach known as the Black-box adversarial attack (Cheng et al., 2018) becomes a viable option. In the black-box attack, the objective is to compromise the model without any prior understanding of its architecture or weight distribution. However, this method demands a substantial number of attempts to be effective. One effective attack within this category harnesses the Rowhammer vulnerability and is termed the Random High-Bit-Flip Attack. In this attack, the attacker randomly selects weights within the DNN model and flips the Most Significant Bit (MSB) or sign bit. Since these attacks are inherently random in nature, they necessitate a substantial number of bit flips to achieve a notable decrease in classification accuracy of the network, as we demonstrate later in this paper.

Following this description of our threat model, we introduce our proposed Outlier Gradient-based Evolutionary (OGE) attack. This attack is characterized by a systematic weight selection process, which unfolds through a three-step approach. The details of each step, which collectively constitute the OGE attack, will be elaborated in subsequent sections.

### 3.2 Gradient-based Layer Selection

A traditional BNN architecture is composed of several convolution and fully connected layers, consisting of millions of parameters. An exhaustive search from this plethora of binary weights would be computationally impractical from an adversarial perspective. To this end, we reformulate the task of obtaining the most vulnerable layers in the BNN by ranking the network loss for top-$k$ gradient bit flips in each layer. In order to obtain the gradients with respect to the latent weights for each mini-batch in the test dataset, we adopted the cost function as a standard categorical cross-entropy loss for $c$ classes, which can be represented as:

$$C = C(Y, g(X, w)) \quad = \quad -\frac{1}{n}\sum_{i=1}^{n}\sum_{r=1}^{c} I(Y_i = r)\log(p_{ir}) = -\frac{1}{n}\sum_{i=1}^{n}\sum_{r=1}^{c} y_i^r \log \hat{y}_i^r \tag{1}$$

where $y_i^r$ is the one-hot coded vectors and $\hat{y}_i^r = p_{ir}$ is the softmax probability for the $i^{th}$ observation to fall in $r^{th}$ class depending on the feature values $X$ and weights $w$. In particular, for classic multinomial logit regression with $c$ classes, $w$ is replaced by the coefficient vector and $p_{ir}$ takes the following form:

$$\hat{y}_i^r = p_{ir} = \mathbb{P}(Y_i = r) = \frac{\exp \beta_r' X_i}{\sum_{s=1}^{c} \exp(\beta_s' X_i)} \tag{2}$$

Here, $X_i$ is the feature vector. However, it has been a common practice to replace the simplistic multinomial logit structure $\beta_S' X_i$ using feature vectors by a suitable neural network structure. In this paper, we consider the same with $g(X, w)$, assuming a DNN structure. This is further extended to accommodate binary weights pertaining to BNNs. First, we demonstrate how to compute the gradients for a proxy real-valued network structure. The corresponding classification probability takes the following form:

$$\hat{y}_i^r = p_{ir} = \mathbb{P}(Y_i = r) = \frac{\exp\{(NN_{W,b}(X_i))_r\}}{\sum_{s=1}^{c} \exp\{(NN_{W,b}(X_i))_s\}} \tag{3}$$

where $(NN_{W,b}(X))_r$ refers to the $r$-th co-ordinate of

$$NN_{W,b}(X) = W_1\sigma_1(W_2 \cdots \sigma_{d-1}(W_d X + b_d)) \cdots)) + b_1 \tag{4}$$

Here $NN_{W,b}(X)$ is a $d-$layer neural network with $c$-dimensional output and $\sigma_i$ are the corresponding activations for the connection between layer $i$ and $i+1$. To calculate the gradient of $C$ in Equation 1 for any entry of the $d$ layers of weight matrices, let us consider a particular weight $w$. Let us fix $i$ and consider $n = 1$ case for the cost function $C$. We write $\hat{y}_i^r = p_{ir} = \exp(z_r)/\sum_s \exp(z_s)$ where $z_r = (NN_{W,b}(X))_r$. Then,

$$\frac{\delta C}{\delta z_r} = -\frac{y_r}{\hat{y}_r}\frac{\delta\hat{y}_r}{\delta z_r} + \sum_{s\neq r}\frac{-y_s}{\hat{y}_s}\frac{\delta\hat{y}_s}{\delta z_r} = -\frac{y_r}{\hat{y}_r}(\hat{y}_r - \hat{y}_r^2) + \sum_{s\neq r}\frac{-y_s}{\hat{y}_s}(-\hat{y}_r\hat{y}_s) = y_r\sum_s y_s - \hat{y}_r = y_r - \hat{y}_r \tag{5}$$

Thus, by chain rule, we can easily compute $\frac{\delta C}{\delta w}$ as following:

$$\frac{\delta C}{\delta w} = \sum_{t=1}^{c}\frac{\delta C}{\delta z_t}\frac{\delta z_t}{\delta w} \tag{6}$$

Here, we omit the detail of computing $\frac{\delta z_t}{\delta w}$, as it heavily depends on network structure involving the specific weight $w$. Note that, for a BNN, the gradient computation is difficult, since the sign function is not even continuous, let alone differentiable. However, we adopt a similar strategy, as delineated in Helwegen et al. (2019), to argue that the computation for the real-valued proxy network suffices all practical purposes.

Upon training the model, gradient calculation of all the trainable convolutional and dense layers with the test data is evaluated. The corresponding weights in each layer are ranked from the highest to the lowest, based on their absolute gradient values. To understand the rationale behind this ranking keeping the layer fixed, consider a specific example of a 3-depth neural net. In light of Equation 6, it suffices to compute $\frac{\delta z_t}{\delta w}$. We rewrite $z$ as:

$$z = W_1 v_1 + b_1, v_1 = \sigma_1(v_2), v_2 = W_2 v_3 + b_2. \tag{7}$$

Now choose $w$ to be one of the entry from $W_2$, say $w_{2,1,1}$. The entries of $\frac{\delta z}{\delta w_{2,1,1}}$ vector looks as following:

$$\frac{\delta z_t}{\delta w_{2,1,1}} = \frac{\delta z_t}{\delta W_1}\frac{\delta\sigma_1(v_2)}{\delta v_2}\frac{\delta v_2}{\delta w_{2,1,1}}. \tag{8}$$

This calculation clearly shows that for all entries of $W_2$ matrix, the other two factors will remain the same; whereas if we choose a weight from a different layer, then the number of components will vary significantly, making two different layers seemingly incomparable. Thus, for our loss/cost gradient computations in Equations 5 and 6, we select the weights having the highest absolute gradient values to flip.

Therefore, in this step, top-$k$ ($k$ is a pre-determined integer value) gradient weight bits of each layer are flipped simultaneously, while keeping all the other layers unaffected. The corresponding effect on the network is evaluated based on loss (Equation 1). Similarly, we calculate these losses for all the other layers independently. Next, the layers are ranked to detect those that contribute the most network loss. This process is executed with different $k$-values and the results are analyzed to detect the most vulnerable layers, as demonstrated later in Section 4.2.1.

### 3.3 Outlier Weight-subset Selection

Once the vulnerable layers are detected, a weight subset is selected from those layers to be provided as input to the next step in the evolutionary framework, as described in Section 3.4. To this end, isolation forest, an outlier detection technique on multivariate data (Liu et al., 2008), is applied on the weights of a layer to find these weights with outlying absolute gradients.

In this step, we iterate the isolation forest applied on the computed $\delta C/\delta w$ at the final $\hat{w}$. For every mini-batch $b_k$ and bit $i$, we compute $\delta C_k/\delta w|_{w=\hat{w}_i'}$ for $i \in \{1, \cdots n\}$,$k \in \{1, \cdots, B\}$ where $\hat{w}_i'$ denotes flipping the

**Algorithm 1** Evolutionary Optimization

**Input**: *Generated Outliers locations* $W = [w_1, \cdots, w_{n_0}]$, *MaxGen*, *P*, *Q*, *R*, *S*
**Output**: *ResultSolution*

1: *SolutionList = EmptyList*
2: **for** *i* in 1....*Q* **do**
3:     *Solution* = choose *P* weights randomly from *W*
4:     Append *Solution* to *SolutionList*
5: **end for**
6: **while** $MaxGen \neq 0$ **do**
7:     *MaxGen = MaxGen − 1*
8:     *LossList = EmptyList*
9:     **for** *i* in 1....*Q* **do**
10:         *Loss = CalculateLoss(Q[i])*
11:         Append *Loss* to *LossList*
12:     **end for**
13:     Sort *SolutionList* in descending order using *LossList*
14:     *SolutionList = SolutionList[: R]*
15:     **for** *i* in 1....*S* **do**
16:         *a1* = Random choice(*SolutionList*)
17:         *a2* = Random choice(*SolutionList*)
18:         $W' = Set\ (a1 + a2)$
19:         *Solution* = choose *P* items randomly from $W'$
20:         Append *Solution* to *SolutionList*
21:     **end for**
22:     *Q = R + S*
23: **end while**
24: *ResultSolution = SolutionList[0]*

$i$-th bit of the final gradient vector $\hat{w} \in \mathbb{R}^n$ and $\delta C_k / \delta w$ denotes the gradient computed for $k-$th mini-batch. Here, $C$ from Equation 1 will be modified by replacing the index belonging in the specific mini-batch. For any bit $x$ and batch $b$, we compute the outlier score $s_{x,b}$ as follows: $s_{x,b} = 2^{-\frac{E(h_b(x))}{c(n)}}$, where $E(h_b(x))$ is the average tree height across the isolation forest for $x$, when the data input is restricted to the batch $b$ and $c(n) = 2H_{n-1} - 2(n-1)/n$, with $H_n$ denoting $n$-th harmonic number. For each fixed bit $x$, we define the aggregated outlier-based frequency score as:

$$sf_x = \#\{b : s_{x,b} \geq s_{y,b} \text{ for all } y \neq x\}. \tag{9}$$

The scores $sf_1, \cdots, sf_n$ are sorted in descending order as:

$$sf_{(n)} > sf_{(n-1)} > \cdots > sf_{(1)}. \tag{10}$$

We denote their corresponding bits as $v_n, v_{n-1}, \cdots, v_1$, where this nomenclature means that $v_n$ is the most vulnerable bit, $v_{n-1}$ is the second most, and so on.

We obtain a specific number of outliers by modifying the contamination parameter in the algorithm. The value of the parameter depends on the size of the weight subset. For instance, if we want to select $n_0$-weights from the total number of weights $N$, then the contamination parameter provided is $k_0 = n_0/N$ and we select $\{v_n, \cdots, v_{n-n_0+1}\}$. Once these vulnerable weights are selected, we proceed to the following step, where this subset of weights is provided as input to the evolutionary algorithm.

## 3.4 Weight Search Space Optimization

Inspired by Darwinian natural evolution, various algorithms have been designed to solve constrained and unconstrained optimization problems with an adaptive heuristic approach based on natural selection (Vikhar, 2016; Slowik & Kwasnicka, 2020). The randomness associated with generating new solutions, coupled with

their qualitative ranking and ability to obtain the acceptable solution within a limited search space makes this evolutionary approach an algorithm of choice over other conventional optimization techniques (Kachitvichyanukul, 2012).

In this paper, we design an evolutionary algorithm for optimizing a search space to contain the most vulnerable binary weight indices. The outlier-set detected in Section 3.3 is taken as the solution space for the optimization. We consider the cross entropy loss in Equation (1) as the fitness function and evaluate it for each generated solution by flipping all its binary weight bits simultaneously.

The proposed evolutionary optimization approach is outlined in Algorithm 1. The subset of selected binary weight indices ($W$) and the variables (defined as follows) $MaxGen$, $P, Q, R$, and $S$ are provided as inputs to the algorithm. First, the evolutionary algorithm generates $Q$ solutions by randomly choosing $P$ weight indices from $W$ for each solution *(lines 1-4)*. Following this, the network loss (fitness value) for each generated solution is evaluated and collected in a list ($LossList$) *(lines 8-11)*. Specifically, the $CalculateLoss$ is computed by fitting the Neural Network $Y, X, NN(W, b)$ as argument in Equation 1, where the weight bits in a specific solution are flipped. Thereafter, the solutions are ranked based on their fitness values and the best $R$ solutions are kept for the next iteration *(lines 13-14)*. These $R$ solutions are then leveraged to generate $S$ new solutions by randomly selecting two solutions ($a1, a2$) from the entire set of $R$ solutions. The weight indices contained in these solutions are pooled together ($W'$), from which $P$ items are randomly selected to generate the new solution *(lines 15-21)*. Here we adopt a traditional two-parent solution in accordance with existing research (Lee, 2002; Sivanandam & Deepa, 2008). Subsequently, the total $R + S$ solutions are used for the next iteration *(line 22)*. The algorithm execution stops after $MaxGen$ iterations, to obtain a solution that contains the most critical binary weights in the network *(lines 6-7, 23)*.

## 4 Experimental Results

### 4.1 Experimental Setup

To assess the effectiveness of our proposed OGE attack framework, we have chosen to employ the widely used Binary Neural Network architecture, BinaryNet (Courbariaux et al., 2016). This network has been trained on three distinct image recognition datasets: Fashion-MNIST, CIFAR10, and GTSRB. The initial classification accuracy obtained upon training with these datasets is as follows: Fashion-MNIST yields a baseline accuracy of 92%, CIFAR10 achieves 80%, and GTSRB reaches an impressive 98%. To gauge the efficiency of the OGE attack scheme, we employ the following key metrics: the number of bit flips ($N_{flip}$) and Relative Accuracy Drop (RAD). The RAD quantifies the classification accuracy of the network after it has been subjected to $N_{flip}$ bit flips. This metric is evaluated in relation to the baseline classification accuracy of the specific network-dataset configuration under consideration. Our objective is to identify the minimum number of weight flips ($N_{flip}$) that maximally degrades the classification accuracy of the BinaryNet architecture, thereby revealing the vulnerability of the model to this form of attack.

### 4.2 Efficiency of the Proposed OGE Attack

#### 4.2.1 Gradient-based Layer Selection

In this experiment, we vary the parameter $k$ in selecting top-$k$ gradient values from each layer of the BinaryNet architecture. Subsequently, by flipping the corresponding binary weights, we observe the network loss to obtain the most vulnerable layers for each network-dataset configuration. The results are outlined in Figure 2. As demonstrated in Figure 2a, top-200 weight flips furnish a cumulative network loss of 0.79 and 0.67 in layers $conv1$ (first convolution layer) and $fc3$ (last fully connected layer), respectively, on the Fashion-MNIST dataset. Except these two, all other layers in the network furnish a loss below 0.5, which is denoted in the figure as *Other Layers*. The network loss increases with $k$; BinaryNet furnishes losses of 2.31 and 1.22 for layers $conv1$ and $fc3$, respectively for top-800 weight bit flips. An identical trend is observed for the other two datasets, as shown in Figures 2b and 2c. Since the layers $conv1$ and $fc3$ exhibit the maximum network loss compared to all other layers in the network, we consider the binary weights from these two layers of the network to obtain the critical weight bits, flipping which will accomplish the adversarial intent.

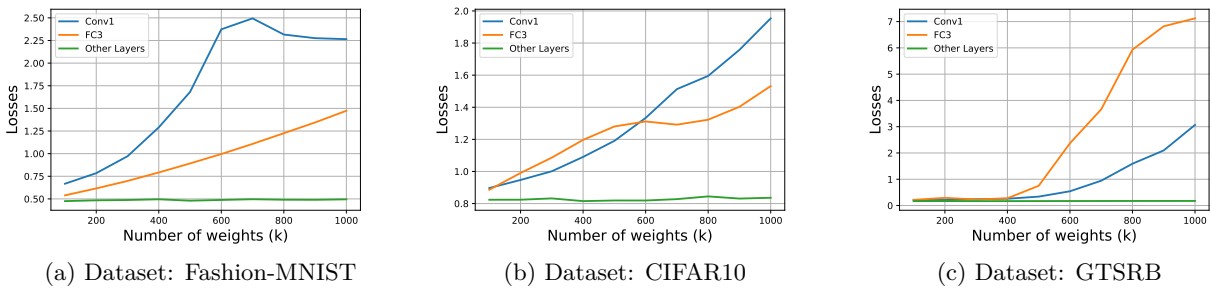

(a) Dataset: Fashion-MNIST      (b) Dataset: CIFAR10      (c) Dataset: GTSRB

Figure 2: Variation of network loss for increasing top-$k$ bit flips across different layers in BinaryNet.

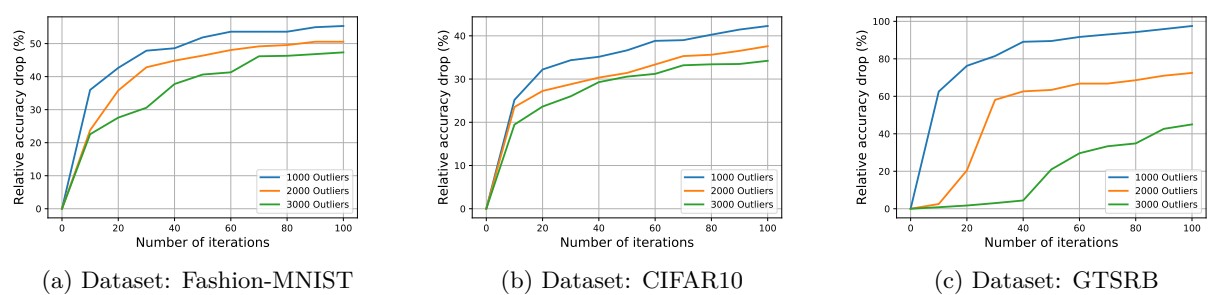

(a) Dataset: Fashion-MNIST      (b) Dataset: CIFAR10      (c) Dataset: GTSRB

Figure 3: Variation in Accuracy w.r.t. iterative generations for different number of outlier gradient weights.

### 4.2.2 Weight-subset Selection using Isolation Forest

In this experiment, we vary the subset of binary weights ($n$) to be selected from the vulnerable layers of the network, obtained in Section 4.2.1. The contamination parameter that determines the cardinality of the subset is varied to obtain the weights with outlier gradients. Correspondingly, this binary weight subset is utilized to analyze the efficiency of the evolutionary search algorithm, by providing this subset as its input solution space. Figure 3a demonstrates the solution optimization space that furnishes the increase in network loss and reduces the accuracy with increasing number of outlier gradient weights on Fashion-MNIST dataset. After 100 iterations, the network exhibits a RAD of 55.32% with 1000 outliers and 300 bit flips. Under identical bit flips, the RAD reaches to 50.56% and 47.34% for 2000 and 3000 outliers, respectively. We observe similar trends for CIFAR10 and GTSRB datasets as well, as represented in Figures 3b and 3c, respectively. The solution with 1000 outliers saturates the search optimization algorithm faster than the remaining outlier sets, and furnishes higher reduction in accuracy compared to the rest. Hence, we choose this configuration to be provided as input to the evolutionary algorithm in the subsequent step.

### 4.2.3 Weight Search Space Optimization

In this section, we first showcase the impact of tuning the evolutionary algorithm, highlighting its capability to optimize the attack strategy. Subsequently, we present the effectiveness of the attack across a range of scenarios involving different numbers of bit flips within the Binary Neural Network (BNN) architecture, illustrating its capacity to induce significant reductions in classification accuracy.

**Tuning the Evolutionary Algorithm:** In this experiment, we vary the maximum number of iterations ($MaxGen$) to execute the evolutionary algorithm, as discussed in Section 3.4. Setting an appropriate $MaxGen$ factor is highly crucial, since early termination of the evolutionary algorithm will result in a non-optimal solution. Fig. 4a demonstrates the variation in accuracy drop with the increase in the number of iterating generations for all three datasets. As outlined in the figure, on Fashion-MNIST dataset, a solution of size containing 200 weight indices from both the $conv1$ and $fc3$ layers, when flipped, furnishes a RAD of 36.85% after 10 iterations, which further increases to 50.9% after 80 iterations. Similarly, for CIFAR10 and GTSRB datasets, we observe an analogous trend in the accuracy drop. The reduction in accuracy

saturates beyond 80 iterations in all three datasets. This motivates us to utilize 80 $MaxGen$ iterations for the evolutionary algorithm.

**Accuracy Drop with Varying $N_{\text{flip}}$:** In this experiment, we vary the number of binary weights ($N_{\text{flip}}$) to be flipped, as obtained from our proposed OGE attack. As shown in the Figure 4b, the accuracy relatively drops 31.5% for 100 $N_{\text{flip}}$ in Fashion-MNIST dataset, which further increases to 64.56% by flipping 400 weights, out of 10.3 million parameters in the model. CIFAR10 furnishes a similar accuracy reduction trend to reach 50.625% RAD for 400 $N_{\text{flip}}$. Similarly, With GTSRB, on flipping 400 binary weights, the BinaryNet furnishes only a minimal classification accuracy of 0.625% with RAD of 99.4%, at which the model can be termed as a random output generator at best. Therefore, our proposed OGE framework is able to subvert the confidence of the system, thereby demonstrating the efficiency of the adversarial bit flip attack.

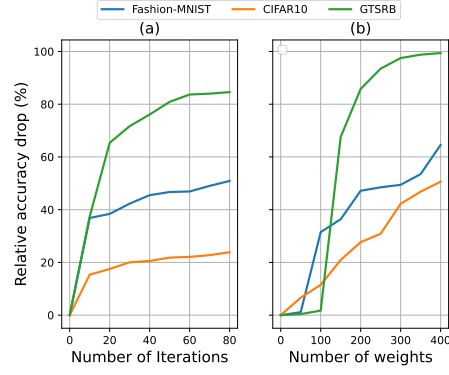

Figure 4: Relative Accuracy degradation for varying number of (a) iterations of the evolutionary algorithm and (b) weight bit flips in the OGE attack.

### 4.3 Comparison with State-of-the-art

In this section, we compare the OGE attack with existing attack strategies, as captured in Table 1. We first compare our strategy with a random bit flip attack that randomly selects weights from the BinaryNet architecture. The corresponding reduction in accuracy is observed for all three datasets. In the case of Fashion-MNIST, as high as 50000 random bit flips furnishes a minuscule 0.2% degradation in accuracy, while our proposed OGE approach, with only 200 bit flips, renders a relative accuracy drop of 54.3%. Identical trends are observed for the other two datasets as well. Therefore, random bit flips are ineffective in reducing the confidence in BNNs, which can be attributed to their inherent robustness arising from limited parameter space.

Table 1: Comparing the efficiency of the proposed OGE attack on BNNs against existing adversarial bit flip-based attacks.

| Dataset | O-Acc(%) | Methods | Affected vulnerable layers | $N_{\text{flip}}$ | PA-Acc(%) | RAD(%) |
|---|---|---|---|---|---|---|
| Fashion-MNIST | 92.07 | Random attack | All layers | 50000 | 91.9 | 0.2 |
| | | BFA (Rakin et al., 2019) | conv1, fc3, conv3 | 500 | 45.2 | 50.8 |
| | | OGE | conv1, fc3 | 200 | 42.1 | 54.3 |
| CIFAR10 | 80.3 | Random attack | All layers | 50000 | 80.05 | 0.3 |
| | | BFA (Rakin et al., 2019) | conv1, fc3, conv2, fc1 | 740 | 38.1 | 52.5 |
| | | OGE | conv1, fc3 | 400 | 39.5 | 50.6 |
| GTSRB | 98.7 | Random attack | All layers | 50000 | 98.5 | 0.2 |
| | | BFA (Rakin et al., 2019) | conv1, fc3, fc1, conv3 | 950 | 42.8 | 56.6 |
| | | OGE | conv1, fc3 | 150 | 31.48 | 68.1 |

The state-of-the-art technique in the domain of un-targeted adversarial bit flipping in DNNs is proposed in Bit flip Attack (BFA) (Rakin et al., 2019). BFA iteratively searches for the most vulnerable layer in the network in a progressive heuristic manner using gradient-based bit flips. Next, on flipping the weight bits in the chosen layer, the model is updated with the modified weights and used for the next iteration. Thus, network loss increases with each loop, which in turn, degrades the overall classification accuracy of the network. We adopted this BFA strategy to attack BNNs, and compare its efficacy with our proposed OGE attack. While BFA requires approximately 500 binary weight flips to obtain a relative accuracy drop (RAD) of 50.8% in the case of Fashion-MNIST, the OGE approach achieves higher accuracy reduction with only 200 bit flips. We observe an identical trend for CIFAR10 dataset as well. The efficiency of our proposed OGE attack is best observed in the case of GTSRB dataset. While BFA requires 950 bit flips to furnish 56.6% relative degradation in accuracy, our OGE attack engenders a much higher RAD of 68.1% with a minimal 150 bit flips in the BinaryNet architecture. Hence, with almost 84% less bit flips, OGE attack achieves even higher degradation in network accuracy, compared to the state-of-the-art, which demonstrates the efficiency

of our proposed attack. The proposed OGE framework furnishes an attack runtime advantage of 3.56× on average over this state-of-the-art BFA method.

### 4.4 Alternate BNN Architectures on ImageNet

In order to evaluate the prowess of our proposed OGE attack framework, we have performed experiments on recent state-of-the-art BNN architectures(as demonstrated in Figure 5), that are trained on ImageNet dataset (Deng et al., 2009). The corresponding baseline classification accuracies are 73.6% for QuickNet, 58.7% for XNORNet, 65.4% for BiRealNet, 61.38% for LQNet (ResNet-18), 62.73% for BinaryResNet18 and 60.6% for BinaryAlexNet. Quicknet demonstrates the highest vulnerability among all the networks, owing to inherent model attributes arising from depthwise separable convolutions and parametric ReLU operations. XNORNet, on the contrary exhibits highest resilience against OGE bit flip attack. This can be attributed to the approximate convolutions using binary operators, that mask the impact of attack on the ensuing classification accuracy of the network. The proposed attack, when compared against BFA and random bit flip attacks, furnishes the highest RAD over the other two attack strategies, thereby demonstrating the efficiency of our proposed approach.

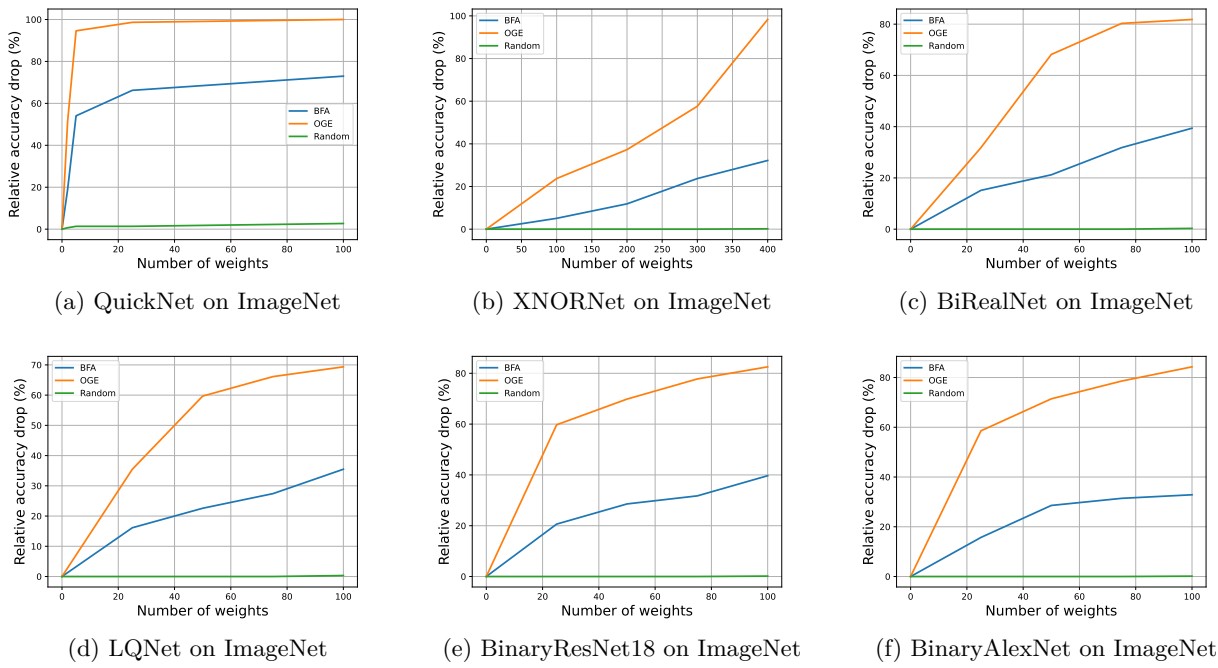

(a) QuickNet on ImageNet   (b) XNORNet on ImageNet   (c) BiRealNet on ImageNet

(d) LQNet on ImageNet   (e) BinaryResNet18 on ImageNet   (f) BinaryAlexNet on ImageNet

Figure 5: Efficiency of OGE attack on state-of-the-art BNN architectures trained on ImageNet.

### 4.5 Evaluations on Novel Architectures

To evaluate the effectiveness of OGE on novel architectures like transformers used in natural language processing (NLP) tasks such as sequence classification, we implement the framework on a Binary precision BERT architecture, termed BinaryBert (Bai et al., 2020). We analyze BinaryBert's performance on MRPC and MNLI datasets (Wang et al., 2018). We assess performance using relative accuracy drop (RAD) and relative drop in F1 score, indicating the model's performance degradation. Higher F1 and RAD drops signify lower model performance. Our attack scheme significantly reduces model accuracy, achieving a 50.8% RAD in Binary-Bert on the MNLI dataset with just 1500 bit-flips (0.000018% of total parameters). Additionally, it consistently yields over 90% relative drop in F1 score, with bit-flips totaling less than 0.000014% of total parameters. This highlights the effectiveness of OGE.

## 4.6 Effectiveness Against State-of-the-Art (SOTA) Defences

We have identified SOTA defenses against bit-flip attacks on deep neural networks (DNNs) (He et al., 2020; Özdenizci & Legenstein, 2022; Wang et al., 2023; Chitsaz et al., 2023). These approaches aim to safeguard full-precision DNNs by either quantizing the model (He et al., 2020; Chitsaz et al., 2023), encrypting the output coding scheme (Özdenizci & Legenstein, 2022), or modifying the model architecture (Wang et al., 2023). However, in the subsequent discussion, we elaborate on why these defenses prove ineffective against our proposed OGE attack framework designed specifically for targeting binary neural networks (BNNs).

The approach outlined in He et al. (2020) suggests using binarization-aware training to convert DNNs into binary form as a defense against bit-flip attacks. However, the OGE framework has shown that binary neural networks can still be effectively attacked with minimal bit-flips, rendering this defense strategy insufficient against our proposed attack. On the other hand, the defense method proposed in Özdenizci & Legenstein (2022) employs an output code-matching strategy to make targeted class attacks more challenging. Yet, the OGE framework takes a broader approach, targeting all classes, making the defense tactic ineffective against our OGE attack. Similarly, the Aegis defense strategy advocated in Wang et al. (2023) proposes a multi-exit approach using multiple internal classifiers. However, if the adversary targets the initial layers before any internal classifier, erroneous activations occur across all classifiers, rendering the Aegis defense ineffective against our OGE strategy. Furthermore, the defense mechanism outlined in Chitsaz et al. (2023) suggests using learned quantization of DNN layers to enhance resilience during inference. Despite this, our study, OGE, reveals that even fully binarized models remain vulnerable to bit-flip attacks. While a defense strategy based on multiple quantization levels may not directly apply to our investigation, considering its implementation is plausible given that targeting a BNN represents the most extreme scenario.

## 4.7 Comparison with Recent Bit-flip Attack Baselines

The research outlined in Chen et al. (2021); Bai et al. (2022) illustrates Trojan attacks on DNNs employing bit-flip methodologies. These attacks pinpoint critical bits within a DNN, which, when attacked by meticulously crafted Trojan patterns, lead to misclassifications into alternative target classes, serving adversarial intentions. In the search for these critical bits within the model, these works rely on the magnitude of high precision (Float16, float32, int8, etc.) weights, which certain optimization techniques exploit to identify these crucial bits. Conversely, BNN weights are solely restricted to +1 or -1, which are represented by a single bit. This renders the optimization techniques utilized in previous studies ineffective in converging and uncovering critical bits within BNNs. Additionally, the attacks discussed in Chen et al. (2021); Bai et al. (2022) are targeted attacks, aiming at input patterns of specific classes or patterns, which causes the overall model accuracy drop to be minimal. However, in our OGE strategy, the attack targets all classes indiscriminately throughout, resulting in substantial drops in model accuracy, and thus demonstrating the efficacy of the attack.

## 5 Ablation Studies

In order to evaluate the efficiency of our OGE attack framework, we performed an extensive ablation study by varying the different design parameters of our algorithm. As summarized in this section, OGE attack furnishes identical trends for various ablation studies on all three datasets – Fashion-MNIST, CIFAR10 and GTSRB. The corresponding results are demonstrated in Figures 6, 7 and 8 respectively.

**Ablation study for layer selection:** In this experiment, we performed ablation study for layer selection, as demonstrated in Figure 7a for CIFAR10 dataset. We observed that, executing OGE attack on selected layers furnishes much higher degradation in classification accuracy, when compared to a scenario where the attack is executed without layer selection. Identical results are obtained for other two datasets as well. This demostrates the importance of layer selection in order to orchestrate an effective bit flip attack on the BNN.

**Ablation study on outlier detection:** In this experiment, we varied the outlier detection approach to select a subset of the binary weights, by considering 4 different techniques – (1) random weight selec-

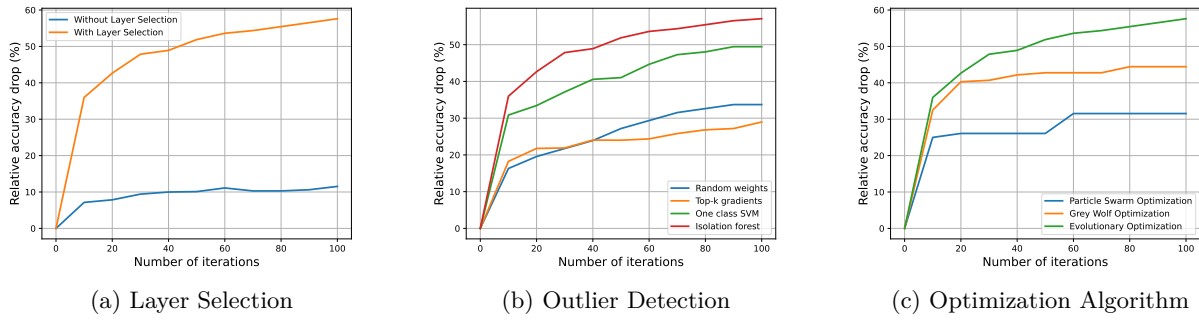

(a) Layer Selection       (b) Outlier Detection       (c) Optimization Algorithm

Figure 6: Ablation studies performed by varying OGE design parameters for Fashion-MNIST dataset.

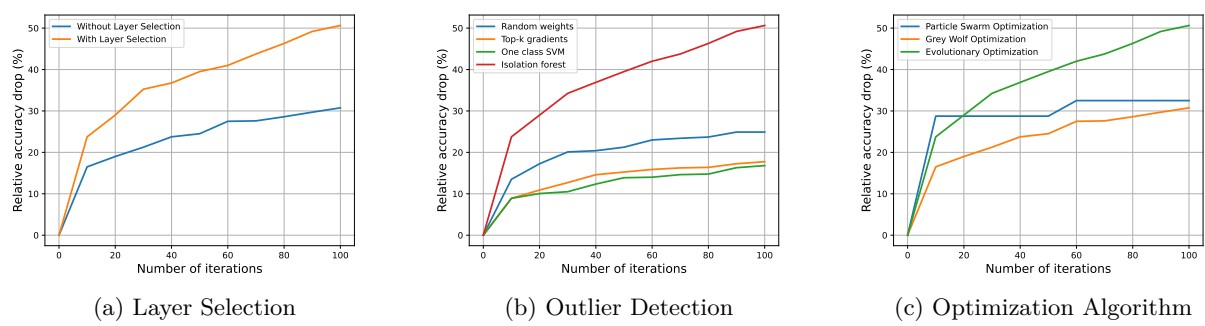

(a) Layer Selection       (b) Outlier Detection       (c) Optimization Algorithm

Figure 7: Ablation studies performed by varying OGE design parameters for CIFAR10 dataset.

tion, (2) weights with top-k gradients, (3) gradient based outlier detection with One-class SVM (OCSVM) with standard *rbf* kernel and, (4) Isolation Forest, that is originally used in the proposed OGE attack.

As demonstrated in Figure 7b, outlier detection with isolation forest exhibits the highest RAD, compared to other weight subset selection techniques. When evaluated on Fashion-MNIST and GTSRB datasets, OGE demonstrated similar results for the isolation forest technique, thereby demonstrating the efficiency of the proposed approach.

Furthermore, we performed experiments with different kernels of OCSVM – (1) *poly*, (2) *sigmoid* and (3) *rbf*, and observed that *rbf* kernel furnishes the highest RAD among other two kernels in OCSVM (as demonstrated in Figure 9 of the Appendix). However, our isolation forest method furnishes even higher degradation in accuracy, when compared to OCSVM method with *rbf* kernel.

**Ablation study with optimization algorithms:** We also performed an ablation study on 3 optimization techniques – (1) Particle Swarm Optimization (PSO) and (2) Grey Wolf Optimization (GWO) along with the proposed (3) evolutionary optimization algorithm, as shown in Figure 7c. The evolutionary

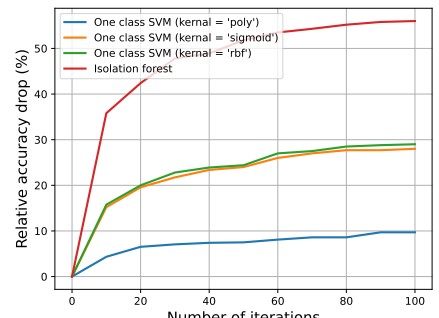

Figure 9: Variation in classification accuracy under OGE attack, for different kernels of OCSVM, which is compared with the performance of isolation forest technique.

approach furnishes the lowest RAD over 100 iterations, compared to PSO and GWO respectively across all three datasets. Similar results are obtained, when we iterated identical experiments on Fashion-MNIST and GTSRB datasets as well.

These comprehensive ablation studies conducted serve to substantiate our design choices for the proposed OGE attack framework, which has proven its efficiency in targeting and compromising the robust binary weights found within a BNN.

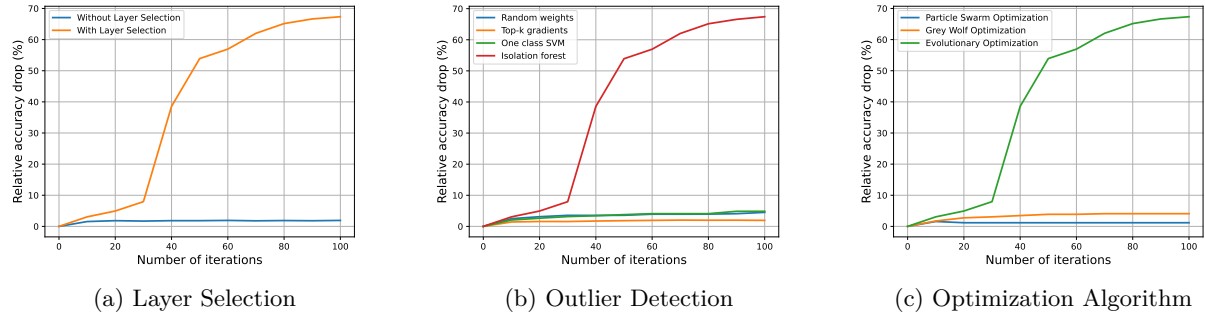

(a) Layer Selection    (b) Outlier Detection    (c) Optimization Algorithm

Figure 8: Ablation studies performed by varying OGE design parameters for GTSRB dataset.

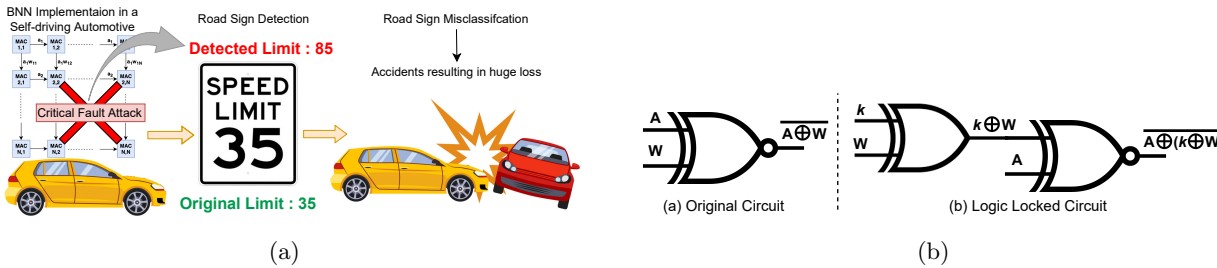

Figure 10: (a) Impact of Bit Flip attack and (b) the proposed defense against OGE attack on BNNs.

## 6    Discussion

Since a BNN demands extremely low amount of resources at the edge, it has the potential to be deployed in the next generation mission-critical applications, *e.g.*, in an autonomous vehicle, as represented in Figure 10a. The decisions ascertained by this network, for instance, detecting a street sign with a speed limit of 35 MPH, enable the automotive to drive as intended. However, under such an adversarial bit-flip attack, the network might incorrectly infer the limit to be 85 MPH, which would direct the vehicle to reach uncontrollable speeds. Since the decisions furnished by such networks are beyond human control, the impact of such attacks can end in catastrophic circumstances, including the loss of human lives. Therefore, it is imperative to explore the vulnerabilities in such extremely low-precision networks and address the manifestation of adversarial attempts that can subvert the confidence of the system.

In order to thwart such attacks, we propose an adaptive defense mechanism. When manifested in a network, this defense strategy aims to jeopardize the underlying strategy of the OGE attack and hence is termed as adaptive. The proposed defense utilizes the concept of XOR-cipher, inspired by the concept of logic locking (Yasin & Sinanoglu, 2017). Once the BNN is trained, a critical subset of vulnerable BNN parameters can be encrypted with XOR-cipher, which, on providing the correct set of keys, will furnish the original accuracy during inference. However, an incorrect key set will exhibit a graceless degradation in accuracy. Therefore, the attacker, without knowing the key, will obtain a network having extremely low baseline accuracy, beyond which, attacking the network further is impractical from an adversarial perspective. Figure 10b demonstrates a high-level representation of this proposed defense strategy. As exhibited in existing research, the multiplication operation in a BNN is accomplished by a logical XNOR operation between the activation (A) and the binary weight (W), thereby resulting in a product of $\overline{A \bigoplus W}$ (Courbariaux et al., 2016). With the implementation of the defense strategy, the weight W will be XOR-ed with a key-bit ($k$), prior to its multiplication with the input activation. Hence, the product obtained can be represented as $\overline{A \bigoplus (W \bigoplus k)}$. Reverse engineering the key is also highly improbable in this scenario; locking $m$ vulnerable binary weights will furnish a probability of $1/2^m$ (*e.g.* approximately $7 \times 10^{-46}$ for 150 encrypted weights). Hence, with this infinitesimal probability, it is almost impossible for the adversary to obtain the correct keys, and subsequently, the accurate set of pre-trained network weights to execute the attack.

**Bottlenecks**: While the defense approach holds the promise of countering the OGE attack, there is a practical hurdle to implementing this defense. The issue stems from the fact that implementing this defense would require provisions at the system level where Binary Neural Networks (BNNs) are executed. Specifically, every position where the weights are mapped would need provisions to encrypt those particular weights. Since BNNs are designed to achieve efficient multiplication operations using XNOR gates, adding XOR gates corresponding to each XNOR gate would substantially increase the area and power overheads necessary for implementing this defense strategy. This addition would nearly double the resources required for processing, significantly compromising the efficiency and effectiveness of such BNNs.

The primary motivation for using BNNs is to make them well-suited for deployment at the edge, where resource constraints and efficiency are paramount. Consequently, implementing a defense strategy that introduces such substantial overhead would undermine the very purpose of deploying BNNs at the edge. This underscores the challenge in developing defense mechanisms for these types of attacks that are both effective and practical, without sacrificing the key advantages that BNNs offer. It highlights the urgent need for innovative defense strategies that can effectively thwart such carefully crafted attacks while remaining compatible with inherently robust BNNs and practical for edge deployment.

## 7    Conclusion

In this work, we propose OGE, an adversarial bit flip attack designed specifically for Binary Neural Networks (BNNs). Our attack aims to disrupt the classification accuracy of these networks, even within their low-precision settings, consequently undermining the system's confidence in its predictions. Our results demonstrate that it's possible to significantly subvert the classification accuracy of a BNN, achieving a Relative Accuracy Drop (RAD) of 68.1% using the OGE methodology. Impressively, this level of disruption can be achieved by flipping as few as 150 weights within a BNN. We achieve this through a systematic approach based on outlier gradients and evolutionary techniques, emphasizing the efficiency of our proposed attack. To defend against such attacks, a potential mitigation strategy could involve the use of XOR-cipher to protect a critical subset of vulnerable bits. However, it's important to note that this defense approach may introduce significant computational and resource overheads, as discussed earlier. Nonetheless, it represents a potential avenue for further exploration in future research, as we continue to grapple with the challenge of securing low-precision networks like BNNs against sophisticated adversarial attacks.

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
