# OpenReview forum: "Bit-by-Bit: Investigating the Vulnerabilities of Binary Neural Networks to Adversarial Bit Flipping"
_TMLR — Accepted by TMLR_

### Review · Reviewer_uVoe · 2024-02-10

**Summary Of Contributions:**

Authors propose an adversarial weight bit-flip attack algorithm for binary neural networks. Former studies so far considered having low-bit quantization as a strong defense in itself, against the existing heuristic attacks on high-precision quantized NNs. This work introduces an attack tailored for the binary case, which challenges the existing belief of weight binarization to be a strong defense against such attacks. Authors' main proposal is the so-called "outlier gradient-based evolutionary (OGE) attack" algorithm, which mainly relies on the information of outlier gradients to restrict its layer/weight search space to detect the set of most vulnerable bits in the BNN. Several experiments are performed across various datasets and binary DNN architectures.

**Audience:**

Yes

**Claims And Evidence:**

Yes

**Requested Changes:**

- An addition (e.g., table) regarding the required time of the algorithm to identify the most vulnerable bits would provide a good comparison, as opposed to BFA (not only Alg. 1, but the complete process starting from gradient-based layer selection).

- In the middle of the Discussion section, suddenly the authors start proposing an adaptive defense mechanism with a logic locked circuit against this vulnerability. Perhaps this should be separated and included in a short additional subsection.

- The paper sufficiently talks about existing weight bit-flip attacks on neural networks, however too little on existing defenses against these challenges. Some notable studies that the authors could discuss in their work:

"Defending and harnessing the bit-flip based adversarial weight attack." CVPR 2020.

"Improving robustness against stealthy weight bit-flip attacks by output code matching." CVPR 2022.

"Aegis: Mitigating Targeted Bit-flip Attacks against Deep Neural Networks." USENIX 2023.

"Training DNNs Resilient to Adversarial and Random Bit-Flips by Learning Quantization Ranges." TMLR 2023.

- Minor comments: The reference citing style of the paper is generally inconsistent and should be re-checked (e.g., use of parantheses in Author et al. style natbib format). There are also several typos and uppercase-lowercase mix-ups here and there.

**Strengths And Weaknesses:**

Strengths:
- The clear contribution of this work is the weight bit-flip attack algorithm that is tailored for the binary quantization case, which is new.
- Experimental results scale to ImageNet level and demonstrate effectiveness of the proposed approach.
- Narrative is clear and the storyline is communicated well.

Weaknesses:
- Scope of the work and the actual utility of a BNN bit-flip attack is highly limited. However I still believe this paper provides a technically valid contribution in its own niche.

---

### Review · Reviewer_AvVk · 2024-02-20

**Summary Of Contributions:**

Binary Neural Networks (BNNs) offer reductions in storage and compute costs compared to traditional Deep Neural Networks (DNNs) by employing ultra-low precision weights. However, their vulnerability to hardware attacks remains a concern, prompting the exploration of adversarial attack paradigms targeting BNNs during deployment to degrade accuracy and undermine confidence in the system. The proposed Outlier Gradient-based Evolutionary (OGE) attack introduces minimal critical bit flips in pre-trained binary network weights, resulting in up to a 68.1% drop in test image misclassification across various datasets, with just 150 binary weight flips out of millions in a BNN architecture.

**Audience:**

Yes

**Claims And Evidence:**

Yes

**Requested Changes:**

- Evaluate the attack against SOTA defenses
- Update the entire paper to the state of research of 2024
     - This includes comparing to the latest BNN attacks
- Provide evaluations on novel architectures (Vision-Transformers, MLP-Mixer, Diffusion models)

**Strengths And Weaknesses:**

Strengths
- The Outlier Gradient-based Evolutionary (OGE) attack appears to be novel to the best of my knowledge.
- The presented attack performances show the vulnerability of BNNs to the proposed attack.
- The introduced minimal critical bit flips to fool a BNN are interesting.

## Weaknesses
- The paper did not evaluate any efficiency against defenses.
- This work is outdated. This work was submitted in 2024 and cites only 3 works from 2023, of which only one is related to BNNs.
- Given the above, I doubt that the authors have truly compared against the state-of-the-art, especially since the works they compare to are from 2019.
- The community has seen various new architectures, Vision-Transformers, MLP-Mixer, Diffusion models, etc. None of these models were tested for the proposed attack.

---

### Review · Reviewer_3iFu · 2024-03-27

**Summary Of Contributions:**

The paper presents an adversarial attack framework called OGE (Optimized Gradient-based Adversarial Bit Flip Attack) specifically designed for Binary Neural Networks (BNNs). The authors propose a multi-stage attack strategy that leverages gradient-based layer selection, weight-subset selection using Isolation Forest, and weight search space optimization through an evolutionary algorithm. The effectiveness of the OGE attack is evaluated on several datasets trained with the different BinaryNet architectures.

**Audience:**

Yes

**Broader Impact Concerns:**

I do not see any broader impact concerns.

**Claims And Evidence:**

Yes

**Requested Changes:**

1. Please discuss the efficiency of the proposed attack.
2. Please evaluate the effectiveness of the proposed attack on some advanced adversarially robust BNNs.
3. Please compare with recent flip attack baselines.

**Strengths And Weaknesses:**

Strengths:
1. The authors conduct experiments on multiple datasets as well as architectures. The comparison with other baselines demonstrates the effectiveness of the proposed attack.
2. The proposed attack is easy-to-follow and the proposed framework seems natural.

Weaknesses:
1. Although the authors claim the efficiency of the proposed attack, it is difficult to see the corresponding analysis of efficiency in the methodology part.
2. The effectiveness of the proposed attack is not evaluated on adversarially robust BNNs, such as [1,2,3].
3. The compared baselines are out-of-date. Some advanced flip attacks should be compared, such as [4,5].


[1]. Improving Robustness Against Stealthy Weight Bit-Flip Attacks by Output Code Matching. CVPR 2022.

[2]. Defending and Harnessing the Bit-Flip based Adversarial Weight Attack. CVPR 2020.

[3]. Training DNNs Resilient to Adversarial and Random Bit-Flips by Learning Quantization Ranges. TMLR 2023.

[4]. ProFlip: Targeted Trojan Attack With Progressive Bit Flips. ICCV 2021.

[5]. Hardly perceptible trojan attack against neural networks with bit flips. ECCV 2022.

---

### Decision · Action_Editor_fNe1 · 2024-05-20

**Recommendation:** Accept with minor revision

**Comment:**

The reviewers raised several points that were addressed in the author's rebuttal. However, the authors did not upload a revision of the paper in accordance with their answers. The authors should upload a revised manuscript that includes additional results and discussions provided in the rebuttal.

**Audience:**

Yes, the submission is interesting for researchers working on adversarial robustness, in particular in combination with binary neural networks.

**Claims And Evidence:**

The claims of the paper are supported by the provided evidence.